# Sulforaphane Potentiates Gemcitabine-Mediated Anti-Cancer Effects against Intrahepatic Cholangiocarcinoma by Inhibiting HDAC Activity

**DOI:** 10.3390/cells12050687

**Published:** 2023-02-22

**Authors:** Fumimasa Tomooka, Kosuke Kaji, Norihisa Nishimura, Takahiro Kubo, Satoshi Iwai, Akihiko Shibamoto, Junya Suzuki, Koh Kitagawa, Tadashi Namisaki, Takemi Akahane, Akira Mitoro, Hitoshi Yoshiji

**Affiliations:** Department of Gastroenterology, Nara Medical University, Kashihara, Nara 634-8521, Japan

**Keywords:** chemoresistance, angiogenesis, EMT, cell cycle arrest, apoptosis

## Abstract

Intrahepatic cholangiocarcinoma (iCCA), the second most common primary liver cancer, has high mortality rates because of its limited treatment options and acquired resistance to chemotherapy. Sulforaphane (SFN), a naturally occurring organosulfur compound found in cruciferous vegetables, exhibits multiple therapeutic properties, such as histone deacetylase (HDAC) inhibition and anti-cancer effects. This study assessed the effects of the combination of SFN and gemcitabine (GEM) on human iCCA cell growth. HuCCT-1 and HuH28 cells, representing moderately differentiated and undifferentiated iCCA, respectively, were treated with SFN and/or GEM. SFN concentration dependently reduced total HDAC activity and promoted total histone H3 acetylation in both iCCA cell lines. SFN synergistically augmented the GEM-mediated attenuation of cell viability and proliferation by inducing G2/M cell cycle arrest and apoptosis in both cell lines, as indicated by the cleavage of caspase-3. SFN also inhibited cancer cell invasion and decreased the expression of pro-angiogenic markers (VEGFA, VEGFR2, HIF-1α, and eNOS) in both iCCA cell lines. Notably, SFN effectively inhibited the GEM-mediated induction of epithelial–mesenchymal transition (EMT). A xenograft assay demonstrated that SFN and GEM substantially attenuated human iCCA cell-derived tumor growth with decreased Ki67^+^ proliferative cells and increased TUNEL^+^ apoptotic cells. The anti-cancer effects of every single agent were markedly augmented by concomitant use. Consistent with the results of in vitro cell cycle analysis, G2/M arrest was indicated by increased p21 and p-Chk2 expression and decreased p-Cdc25C expression in the tumors of SFN- and GEM-treated mice. Moreover, treatment with SFN inhibited CD34-positive neovascularization with decreased VEGF expression and GEM-induced EMT in iCCA-derived xenografted tumors. In conclusion, these results suggest that combination therapy with SFN with GEM is a potential novel option for iCCA treatment.

## 1. Introduction

Intrahepatic cholangiocarcinoma (iCCA) is the second most common hepatic malignancy arising from intrahepatic bile duct epithelium [1]. The prognosis of iCCA is poor because of early local invasion; metastasis to the liver, lymph nodes, and other organs; and insufficient early diagnosis [1,2]. Currently, only a small number of patients with iCCA can undergo curative resection. Meanwhile, the treatment landscape of unresectable advanced iCCA has primarily been limited to chemotherapy. At present, the first-line chemotherapy for unresectable iCCA is gemcitabine (GEM) and cisplatin (CDDP) based on the ABC-02 study, and second-line chemotherapy includes 5-fluorouracil, folinic acid, and oxaliplatin (FOLFOX) based on the ABC-06 study [3,4]. However, median overall survival, even with these options, is limited to just one year [4]. Additionally, combination treatment with multiple anti-cancer drugs often results in severe adverse effects [3,4]. Currently, several approaches are employed to find novel combinatory treatments with standard chemotherapeutic drugs, including GEM for other types of cancer, such as the highly aggressive diffuse malignant peritoneal mesothelioma and pancreatic ductal adenocarcinoma [5,6]. Likewise, there is an urgent need to identify novel therapeutic targets for iCCA with less adverse event profiles by combining GEM.

Histone deacetylases (HDACs) play a key role in epigenetically regulating the expression and activity of various factors relevant to carcinogenesis and cancer development [7,8]. HDACs comprise a family of enzymes categorized into four classes in humans based on their homology to yeast HDAC analogs: classes I (HDAC1, 2, 3, and 8), II (HDAC4, 5, 6, 7, 9, and 10), III (sirtuins), and IV (HDAC11). Class I, II, and IV HDACs require zinc-dependent cofactors for their enzymatic activity, and class III HDACs require nicotinamide adenine dinucleotide-dependent cofactors [7,9]. Histone acetyltransferases (HATs) catalyze the transfer of an acetyl group from acetyl coenzyme A, while HDACs remove acetyl groups from histones and organize a non-permissive chromatin conformation, leading to interference with the transcription of cancer-related genes [10]. Aberrant HDAC activity leads to diverse transcriptional gene regulation relevant to cancer cell differentiation, angiogenesis, proliferation, apoptosis, migration, and metastasis [10,11]. HDAC activity represses p53 and BAX and induces BCL-2, which promotes cell cycle progression and regulates apoptosis in cancer cells [12,13]. Morine et al. have reported that intratumor HDAC expression is positively correlated with HIF-1α, a stimulus factor for local hypoxia and increased angiogenesis in resected iCCA tissues [14]. Thus, HDAC inhibitors have the potential to thwart cell growth, accelerate differentiation, and induce apoptosis, and they have been proposed as novel therapeutic options for a variety of malignancies, including iCCA [10,11,15].

Sulforaphane (SFN), an isothiocyanate cleavage product of glucoraphanin, can be obtained from damaged cruciferous vegetables such as broccoli, cauliflower, cabbage, and Brussels sprouts [16]. SFN possesses anti-oxidative properties with multiple pharmacological actions, including anti-diabetic and anti-microbial effects [17,18]. Remarkably, SFN has been suggested to display anti-cancer and chemopreventive properties by inhibiting HDAC activity and epigenetically modifying the expression of critical cytoprotective genes involved in the regulation of the cell cycle and apoptosis [19,20]. A recent report revealed that SFN could inhibit total HDAC activity in cancer cells [19]. Moreover, recent findings indicated that SFN augments the response to several carcinostatic agents by enhancing the sensitivity and suppressing the resistance of cancer cells to these agents [21,22].

Based on these findings, the present study investigated the combinatorial effect of SFN and GEM on human iCCA cell growth and malignant potential using iCCA-derived murine xenograft models.

## 2. Materials and Methods

### 2.1. Compounds and Cell Culture

d,l-sulforaphane (1-isothiocyanate-4-methylsulphinylbutane, purity ≥ 98%) was purchased from Toronto Research Chemicals Inc. (Toronto, ON, Canada), and gemcitabine (2′-deoxy-2′,2′-difluorocytidine, purity ≥ 98%) was purchased from Tokyo Chemical Industry Co., Ltd. (Tokyo, Japan). Two human iCCA cell lines, HuCCT-1 (cat: JCRB0425) and HuH28 (cat: JCRB0426) were obtained from the Japanese Collection of Research Bioresources Cell Bank (Osaka, Japan). These cells were cultured in RPMI-1640 (Nacalai Tesque, Inc., Kyoto, Japan) supplemented with 10% fetal bovine serum (FBS) and 1% ampicillin/streptomycin. The primary human biliary epithelial cell line (HIBEpiC, cat: #5100) was purchased from ScienCell Research Laboratories, Inc. (Carlsbad, CA, USA). HIBEpiC cells were cultured in Epithelial Cell Medium (ScienCell Research Laboratories) supplemented with 2% FBS and 1% epithelial cell growth supplement (ScienCell Research Laboratories), and 1% ampicillin/streptomycin. The cells were grown at 37 °C in a 5% CO_2_ atmosphere.

### 2.2. Human iCCA Xenograft Model

Six-week-old male athymic nude mice (BALB/cSlc-nu/nu) (Japan SLC, Inc., Shizuoka, Japan) were housed in stainless steel mesh cages (2/cage) under controlled conditions (temperature: 23 ± 3 °C, relative humidity: 50 ± 20%, 10–15 air changes/h, illumination: 12 h/d). The animals were allowed tap water access ad libitum throughout the study period. Eighty mice were used in total for the xenograft assay, and tumor inoculation was performed as described [23]. Briefly, a million cells were suspended in 200 μL of medium containing Matrigel (Corning, Tewksbury, MA, USA; 1:1), and the same type of million cells was inoculated subcutaneously into the bilateral flanks of each mouse. Tumors were measured with a caliper, and the tumor volume was calculated using the following formula:(1)$$\frac{1}{2}\left[{\left(Width\right)}^{2}\times Length\right]$$

Five days after inoculation, mice were orally administered with SFN (50 mg/kg/day) or intraperitoneally injected with GEM (100 mg/kg) twice a week or concomitant administration [23,24] (*n* = 10). Saline solution was equivalently given to the vehicle group (*n* = 10). The condition and health of mice were monitored daily after the injection of tumor cells, and all mice were sacrificed 30 days after drug administration under anesthesia with barbiturate overdose (intravenous injection, 150 mg/kg pentobarbital sodium). All the animal procedures were performed as per the recommendations of the Guide for Care and Use of Laboratory Animals (National Research Council, Washington, DC, USA). The study was approved by the animal facility committee of Nara Medical University (Authorization number: #12853).

### 2.3. Detection of HDAC/HAT Activity and Total Histone H3 and H4 Acetylation

HuCCT-1 and HuH28 cells were treated with different concentrations of SFN (0–80 μM) or GEM (0–10 μM) for 3 h. To measure HDAC activity, nuclear extracts were obtained from cultured cells or 20 mg of subcutaneous tumor samples using an EpiQuik™ Nuclear Extraction Kit (Epigentek, Farmingdale, NY, USA) according to the manufacturer’s protocol. HDAC activity was measured in 10 μg of nuclear extract using an EpiQuik™ HDAC activity/inhibition assay kit (Epigentek) according to the manufacturer’s instructions. HAT activity was also measured in 10 μg of nuclear extract from cultured cells using an EpiQuik™ HAT activity/inhibition assay kit (Epigentek) according to the manufacturer’s instructions.

To detect total histone H3 and H4 acetylation, histone extracts were obtained from cultured cells using an EpiQuik™ Total Histone Extraction Kit (Epigentek). Histone H3 and H4 acetylation was detected in 100 ng of histone extract using an EpiQuik™ Total Histone H3 Acetylation Detection Fast Kit and an EpiQuik™ Total Histone H4 Acetylation Detection Fast Kit (Epigentek) according to the manufacturer’s instructions, respectively. 

Dimethyl sulfoxide (DMSO, Nacalai Tesque, Inc.) was used as a vehicle, and HDAC activity and total histone H3 acetylation in cells treated with SFN and/or GEM were measured relative to that in the vehicle treatment group.

### 2.4. Histone H3 Peptide Array

HuCCT-1 and HuH28 cells were treated with a concentration of dimethyl sulfoxide (DMSO, Nacalai Tesque, Inc.) as a vehicle or SFN (20 μM) for 3 h. Nuclear extracts were obtained from cultured cells using an EpiQuik™ Nuclear Extraction Kit (Epigentek, Farmingdale, NY, USA) according to the manufacturer’s protocol. To profile the binding specificity of histone H3 acetylation, we used a Pre-Sure™ Histone H3 Peptide Array ELISA Kit (Epigentek) according to the manufacturer’s instructions and previous report [25]. Total nuclear extracts were diluted to 1 ug/mL, added to the array plate and incubated for 2 h at room temperature. Histone H3 Acetylation Antibody Panel Pack I and Pack II (Epigentek) were applied as primary antibodies to detect the binding of H3 lysines (K)9, K14, K18, K27, K36, K56, and K79 to histone peptides. Following the incubation with primary antibodies at 37 °C for 60 min, samples were incubated with secondary antibodies (0.4 μg/mL) at room temperature for 60 min and then developed at room temperature for 10 min away from light. Arbitrary units were measured at the absorbance (450 nm) to represent the relative levels of binding specificity and calculated the ratio to the values of Veh treatment.

### 2.5. Cell Viability Assay and Analysis of Cytotoxic Synergy

HuCCT-1 and HuH28 cells were seeded in 96-well plates with RPMI-1640, as previously described. Then, the cells were treated with different doses of SFN (0–80 μM) or GEM (0–10 μM) for 24 h. Cell viability was evaluated by The Premix Water-Soluble Tetrazolium salt (WST)-1 Cell Proliferation Assay system (Takara Bio, Kusatsu, Japan) according to the manufacturer’s protocol. Cell viability was assessed relative to that in the groups without each treatment, and half-maximal inhibitory concentration (IC_50_) was calculated via non-linear regression analysis using GraphPad Prism 9 ver 9.3.1 (GraphPad Software Inc., La Jolla, CA, USA) [26].

To assess the synergy of drug combinations, a combination index (CI) was calculated by the Chou-Talalay method using CompuSyn software version 1.0 (ComboSyn, Inc., New York, NY, USA) [27]. CI gives a quantitative definition of synergism (CI < 1), additive effect (CI = 1), and antagonism (CI > 1). For this purpose, the cells were also exposed to different concentrations of SFN and GEM for 24 h.

### 2.6. Statistical Analysis

All data were statistically analyzed using GraphPad Prism 9 software. Data were indicated as the mean ± standard deviation (SD). Means were compared between two groups by Student’s *t*-test. A one-way analysis of variance followed by Bonferroni’s post hoc test was performed for multiple comparisons. *p* < 0.05 denoted a statistically significant difference.

Additional methods can be found online in the Appendix A.

## 3. Results

### 3.1. SFN Attenuates HDAC Activity and Promotes Histone H3 Acetylation in Human iCCA Cells

We initially examined the effects of SFN and GEM on HDAC activity, moderately differentiated HuCCT-1 cells and undifferentiated HuH28 cells. As presented in Figure 1A,B, SFN concentration-dependently reduced HDAC activity in both HuCCT-1 and HuH28 cells, and the suppressive effects were significant at concentrations exceeding 20 μM. On the contrary, GEM did not alter HDAC activity in these cells at any concentration (Appendix A). Meanwhile, the activity of HAT, a key enzyme that acetylate conserved lysine amino acids on histone proteins, was significantly increased by treatment with SFN at concentrations exceeding 20 μM (Figure 1C,D). Reflecting the reduced HDAC activity, total histone H3 acetylation in both iCCA cell lines was increased by treatment with SFN in a concentration-dependent manner (Figure 1E,F). On the other hand, total histone H4 acetylation was not altered by treatment with SFN (Figure 1G,H). We further determined the acetylation patterns of specific lysine residues on the tails of histone H3 modified by treatment with SFN in iCCA cells. To this end, nuclear protein extracts of both HuCCT-1 and HuH28 cells treated with SFN (20 μM) were utilized for the identification of the acetylation profile of H3 on K9, K14, K18, K27, K36, K56 and K79. As shown in Figure 1I,J, treatment with SFN particularly increased acetylation as compared to vehicle treatment at H3K9 and H3K27 in both HuCCT-1 and HuH28 cells. Moreover, we confirmed that SFN did not affect HDAC activity in normal HIBEpiC cells (Appendix A).

### 3.2. SFN Has a Synergistic Effect with GEM-Mediated Cell Growth Inhibition in Human iCCA Cells

Next, we investigated the impact of SFN and GEM at different concentrations on the viability of HuCCT-1 and HuH28 cells. As presented in Figure 2A,B, SFN efficiently ameliorated HuCCT-1 and HuH28 cell viability with IC_50_ values of 27.4 and 34.2 μM, respectively. Meanwhile, GEM attenuated the viability of both cell lines (IC_50_ of 0.57 μM for HuCCT-1 cells and 0.71 μM for HuH28 cells) as expected (Figure 2A,B). Both agents did not affect the cell viability of normal HIBEpiC at this range of concentrations (Appendix A).

Based on the optimal concentrations, we calculated CI to evaluate whether the cytotoxic effect of combined SFN and GEM against iCCA cell growth is synergistic against iCCA cell growth. As shown in Figure 2C, the CI values calculated by CompuSyn software were 0.552/0.624/0.497, 0.228/0.406/0.183, and 0.227/0.136/0.119, when SFN (6.8 μM) and GEM (0.14/0.28/0.57 μM), SFN (13.7 μM) and GEM (0.14/0.28/0.57 μM), and SFN (27.4 μM) and GEM (0.14/0.28/0.57 μM) were concurrently administered to HuCCT-1 cells, respectively. These CI values were less than 1.0, indicating that the combination of SFN with GEM has synergistic effects on suppressing the viability of HuCCT-1 cells. Combination treatment with SFN and GEM also exerted a synergistic effect against HuH28 cell viability. The CI values were 0.699/0.738/0.628, 0.519/0.714/0.487, and 0.323/0.414/0.299 when SFN (8.5 μM) and GEM (0.17/0.35/0.71 μM), SFN (17.1 μM) and GEM (0.17/0.35/0.71 μM), and SFN (34.2 μM) and GEM (0.17/0.35/0.71 μM) were concurrently cultivated (Figure 2D). 

We further confirmed that the combination of SFN and GEM at IC_50_ significantly suppressed the proliferative activity of HuCCT-1 and HuH28 cells in a time-dependent manner (Figure 2E,F).

### 3.3. SFN Induces G2/M Arrest and Promotes Apoptosis in Human iCCA Cells

SFN-mediated HDAC inhibition has been reported to enhance histone acetylation and derepress p21 and BAX gene expression, resulting in the induction of cell cycle arrest/apoptosis in several types of cancer cells [19,28,29]. Based on these findings, we examined the effects of SFN on the cell cycle/apoptosis and the expressions of associated genes, including these key targets in human iCCA cells. As presented in Figure 3A,B, SFN or GEM significantly blocked both HuCCT-1 and HuH28 cells in the G2 phase compared to the effects of the vehicle, and the drugs in combination had significantly stronger effects than either agent alone. The mRNA expression of *CDKN1A* and *BAX* were significantly increased by treatment with SFN as compared to vehicle treatment in both HuCCT-1 and HuH28 cells (Figure 3C). Treatment with SFN as well as GEM upregulated p21 and p-Chk2 and downregulated p-Cdc25C at the protein level, corresponding to the induction of G2/M arrest, in both cell lines (Figure 3D). SFN- or GEM-treated HuCCT-1 and HuH28 cells also increased pro-apoptotic BAX expression and decreased anti-apoptotic BCL-2 expression (Figure 3E). In both cell lines, combination treatment augmented the upregulation of BAX compared to the effect of every single agent (Figure 3E). SFN and GEM further enhanced the cleavage of caspase-3, reflecting the induction of cell apoptosis in both HuCCT-1 and HuH28 cells (Figure 3F).

### 3.4. SFN Inhibits Cancer Cell Invasion, Migration, Angiogenic Activity, and Epithelial-Mesenchymal Transition (EMT) in Human iCCA Cells

Next, we investigated the effects of SFN and GEM on malignant potential, including cell invasion, migration, angiogenic activity, and EMT in human iCCA cells. First, the effects of both agents on the invasiveness of HuCCT-1 and HuH28 cells were evaluated using a Matrigel invasion assay. Either drug alone significantly reduced the invasiveness of both HuCCT-1 and HuH28 cells (Figure 4A,B). It was noteworthy that concomitant treatment with SFN and GEM extensively reduced cell invasion to less than 20% of the control, exceeding the effects of each drug (Figure 4A,B). Correspondingly, cell migration was also suppressed by treatment with SFN or GEM in both iCCA cells (Figure 4C). Moreover, combination treatment enhanced the suppressive effect of every single agent (Figure 4C). We next examined the effects of SFN on the angiogenic activity of iCCA cells. Treatment with SFN significantly reduced the mRNA expression of pro-angiogenic markers, including *VEGFA*, *VEGFR2*, *HIF1A*, and *NOS3* in both HuCCT-1 and HuH28 cells (Figure 4D,E). Moreover, we assessed the effects of both agents on the EMT status. There were differences in EMT-related markers between HuCCT-1 and HuH28 cells, which have different levels of differentiation. Specifically, HuCCT-1 cells, which are moderately differentiated, exhibited higher expression of the epithelial markers *CDH1* and *KRT19* and lower expression of the mesenchymal markers *CDH2*, *VIM*, *MMP2*, and *MMP9* than undifferentiated HuH28 cells, consistent with a previous report (Figure 4F) [30]. As presented in Figure 4G,H, treatment with GEM downregulated the epithelial markers and upregulated the mesenchymal markers in HuCCT-1 and HuH28 cells, indicating the EMT progression. Notably, SFN efficiently inhibited the GEM-induced progression of EMT in iCCA cells (Figure 4G,H). 

### 3.5. SFN Potentiates the GEM-Mediated Reduction of the Human iCCA-Derived Xenograft Tumor Growth

Given the suppressive effects of SFN and GEM on human iCCA cell growth, the anti-cancer property of both agents was examined using iCCA-derived xenograft models (Figure 5A). Initially, we determined the experimental dose of SFN for in vivo study. As SFN is also known to exert anti-oxidative effects via Nrf2 activation, we measured the hepatic mRNA levels of anti-oxidative markers in nude mice treated with different doses of SFN to identify a dose that could exert bioactivity in mice [19]. As presented in Appendix A, oral administration of SFN for four weeks increased the hepatic mRNA expression of *Hmox1*, *Nqo1*, and *Gstm3* in a dose-dependent manner even with concomitant GEM treatment (100 mg/kg twice a week), and we identified 50 mg/kg/day as the minimal dose that significantly induced these anti-oxidative genes.

Based on this result, we employed 50 mg/kg/day as the experimental dose for the xenograft assay. Serological assessments revealed that this dose of SFN did not cause hepatocellular, biliary, or renal damage in mice, even when used together and combined with GEM (Appendix A). In mice treated with either SFN (50 mg/kg/day) or GEM (100 mg/kg twice a week), the HuCCT-1 and HuH28-grafted subcutaneous tumor growth was markedly attenuated (Figure 5B). After treatment for 30 days, the subcutaneous tumor volumes and weights were significantly reduced in mice treated with either SFN or GEM relative to the findings in vehicle-treated mice (Figure 5B,C). Notably, concomitant treatment with both agents significantly potentiated their inhibitory effects on tumor growth relative to every single agent (Figure 5B,C). H&E staining illustrated that the viable cancer area in resected subcutaneous tumors was decreased by treatment with SFN and GEM (Figure 5D). We confirmed that the utilized dose of SFN effectively decreased HDAC activity in the resected subcutaneous tumor tissues to less than 60% of that in the vehicle group (Figure 5E).

### 3.6. SFN Suppresses Cell Proliferation and Induces Apoptosis in Human iCCA-Derived Xenograft Tumors

We next quantitatively investigated cancer cell viability in xenograft tumors derived from HuCCT-1 and HuH28 cells (Figure 6 and Appendix A, respectively). In HuCCT-1–derived xenograft tumors, Ki67-positive cancer cell proliferation was attenuated by each drug alone, and the effect was enhanced by using the drugs in combination (Figure 6A). Quantitative analysis revealed the potent reduction of proliferative cells to less than 20% of the control level by combination treatment (Figure 6B). Treatment with SFN and GEM significantly increased the nuclear expression of p21 and cytosolic expression of p-Chk2 and conversely decreased the expression of p-Cdc25C (Figure 6C–E). These findings aligned with the observation of G2/M arrest following treatment with SFN and GEM in iCCA cells. Meanwhile, we found that TUNEL-positive cell apoptosis was simultaneously increased by treatment with SFN and GEM in HuCCT-1–derived xenograft tumors (Figure 6F,G). Notably, the effects of SFN and GEM on intratumor cancer cell viability were also observed in the HuH28-derived xenograft tumors (Appendix A) 

### 3.7. SFN Attenuates Intratumor Angiogenesis and GEM-Mediated EMT in Human iCCA-Derived Xenograft Tumors

Moreover, the effects of SFN and GEM on malignant potential, including pathological angiogenesis and EMT in the xenograft tumors, were examined according to the findings of the in vitro study. As presented in Figure 7A, CD34-positive neovascularization in xenograft tumors derived from both HuCCT-1 and HuH28 was significantly reduced by treatment with SFN. However, these anti-angiogenic effects were not observed in GEM-treated mice (Figure 7A). The semi-quantitative analysis illustrated that the number of new CD34-positive intratumor vessels was decreased by 50% in SFN-treated mice compared to that in vehicle-treated mice (Figure 7B). In parallel with reduced neovascularization, the intratumor expression of *VEGFA* and *VEGFR2* was decreased in SFN-treated mice (Figure 7C). Regarding EMT-related markers, we found that the intratumor mRNA expression of epithelial markers (*CDH1* and *KRT19*) was decreased in GEM-treated mice, and this effect was efficiently inhibited by SFN treatment (Figure 7D). In contrast, treatment with SFN considerably attenuated the GEM-mediated increases in mesenchymal markers (*CDH2*, *VIM*, *MMP2*, and *MMP9*, Figure 7D). Moreover, the effects of both agents on EMT-related markers were similarly observed at the protein levels (Appendix A). These findings indicate that SFN ameliorated resistance to GEM by suppressing tumor angiogenesis and EMT in iCCA cells. 

## 4. Discussion

This study first demonstrated that SFN, a phytochemical isothiocyanate agent, effectively augmented the inhibitory effect of GEM on iCCA growth. Our results demonstrated that SFN exerted multifunctional properties against the malignant potential of iCCA, including anti-proliferative, pro-apoptotic, anti-invasive/migratory, anti-EMT and anti-angiogenic effects. As the functional mechanism underlying these effects of SFN, we suggested the inhibitory action on HDAC activity as well as the inductive action on HAT activity leading to enhancement of histone H3, particularly H3K9 and H3K27 acetylation. Previous studies reported that the dysfunction of HDAC enzymes and altered acetylation status is relevant to the growth and malignant progression of CCA, including iCCA, and several HDAC inhibitors have displayed suppressive effects on iCCA [14,15,31,32,33]. For instance, chidamide, an HDAC inhibitor, has been reported to exert antitumor activities in iCCA by promoting HDAC3-mediated forkhead box O1 acetylation [15]. Another report has shown that peanut testa possessing HDAC inhibitory activity induces apoptosis in iCCA cells [34]. Moreover, a recent report has demonstrated that SFN increases HAT activity in human malignant melanoma cells [35]. These pieces of evidence support the possible involvement of epigenetic modification in the SFN-mediated anti-cancer property against human iCCA cells in our study. On the other hand, we found that SFN did not affect histone H4 acetylation. The present study did not identify a pharmacological mechanism to explain the differential effect of sulforaphane on H3 and H4 acetylation. Thus, a detailed analysis is required in the future.

The present study primarily elucidated that SFN effectively suppressed cell proliferation in both moderately differentiated and undifferentiated iCCA lines. Several reports have shown that SFN-mediated anti-cancer effects were involved in an increase of acetylated histone H3 specifically associated with the promoter region of the p21 and BAX genes in cancer cells [36,37]. Consistently with this evidence, our results showed that SFN increased p21 expression leading to the phosphorylation of Chk2 and dephosphorylation of Cdc25C, and consequently, it blocked cell cycle progression in the G2/M phase in human iCCA cells. SFN also upregulated BAX expression, downregulated BCL-2 expression, and suppressed the cleavage of caspase-3, indicating the activation of the mitochondrial apoptotic pathway. It is known that H3K9ac and H3K27ac are highly correlated with transcriptional activation [38]. Therefore, we hypothesized that SFN-mediated HDAC inhibition possibly promoted the transcriptional activity of p21 and BAX by binding to both genes, enhancing the binding of active modification of histones such as H3K9ac and H3K27ac to regulate the expressions of both genes, thereby suppressing cell proliferation and augmenting cell apoptosis. However, further investigation is necessary to clarify the histone acetylation in the promoters of p21 and BAX, as well as the possible targets downstream of decreased HDAC in SFN-treated human iCCA cells.

Of note, present results showed that a combination of SFN and GEM was likely to ameliorate xenograft iCCA tumor progression more potently than cultured cell growth. This discrepant finding is suggested to be attributable to the impact of SFN on other malignant phenotypes, including cell invasion, angiogenesis, and EMT. The combination treatment of SFN and GEM effectively suppressed cell invasion and migration at the doses with tolerable cytotoxicity, consistent with the results from Wang et al. that co-treatment of iCCA cells with several types of HDAC inhibitors (trichostatin A and valproic acid) and GEM inhibited cell invasion, migration [33]. Moreover, we found that treatment with GEM accelerated EMT, as indicated by the downregulation of epithelial markers and upregulation of mesenchymal markers in both cultured iCCA cells and xenografted tumors. It was noteworthy that SFN effectively inhibited GEM-induced EMT in both iCCA cell lines. Among the malignant phenotypes, EMT has recently gained attention as a potential mechanism of chemoresistance because of its ability to promote the acquisition of cancer stemness and confer resistance to chemotherapy [39]. Indeed, resistance to GEM in iCCA is also associated with EMT phenotype and cancer stem-like properties in the tumor [40]. EMT is regulated by epigenetic changes, including histone modifications, and HDAC inhibitors are considered to modify EMT-related factors’ expression depending on the cancertype [41]. Meanwhile, SFN is reported to inhibit EMT in several cancer cell types by molecular mechanisms independently of histone modification [42,43,44]. Recent studies illustrated that SFN could suppress the EMT in lung cancer cells by inhibiting the GSK3β/β-catenin pathway and ERK5 activation [42,45]. Li et al. also demonstrated that SFN-mediated inhibition of the sonic hedgehog–GLI pathway resulted in the suppression of EMT in pancreatic cancer [46]. These findings evoke a hypothesis that the SFN-mediated suppression of EMT in iCCA cells involves mechanisms beyond the inhibition of HDACs. Thus, additional analyses are needed to clarify the underlying mechanism.

Furthermore, tumor-associated angiogenesis and VEGF expression are known to be correlated with iCCA cancer progression, metastasis, and prognosis [47]. A previous observational study found that VEGF was expressed in 53.8% of 106 patients with iCCA, and it was significantly associated with intrahepatic metastasis [48]. Notably, SFN exerts anti-angiogenic effects by inhibiting hypoxia-induced HIF-1α and VEGF expression in several cancers, including prostate, colon, and liver cancers [49,50,51]. Moreover, SFN has been demonstrated to directly suppress proliferation, tubular formation, and matrix metalloproteinase production in vascular endothelial cells [52]. We substantiated that SFN reduced the expression of pro-angiogenic genes such as VEGFA, VEGFR2, HIF-1α, and eNOS in iCCA cells and attenuated CD34-positive neovascularization in xenografted tumors. As the inhibition of angiogenesis has been reported to abolish chemoresistance to GEM, this anti-angiogenic property of SFN is potently associated with the augmentation of GEM-mediated anti-cancer effects on iCCA [53].

The empirical results reported in this study should be considered in light of some limitations. First, we demonstrated that the combination of SFN and GEM synergistically augmented the anti-cancer effect on human iCCA cells by calculating the CI. However, our study did not fully elucidate the pharmacological interaction between both agents to explain this synergistic effect. Although the inhibition of GEM-induced EMT by SFN is likely to be associated with this synergy, we will probably need further detailed research by comprehensive molecular profiling. Second, although we defined the dose of SFN (50 mg/kg/day) for the in vivo study, optimization is performed by evaluating the anti-oxidative property of SFN in the liver. We confirmed that this dose efficiently reduced HDAC activity in the xenografted tumors. Additionally, we observed that the doses of SFN and GEM did not cause hepatic, biliary, or renal toxicity in mice. Thus, these doses are assumed to be within the tolerance range for in vivo experiments. Third, the current first-line chemotherapy for iCCA is based on GEM and CDDP. Therefore, additional study is necessary to substantiate the enhanced efficacy of GEM and CDDP in combination with SFN in the latest clinical setting.

In summary, the present study demonstrated that combination with SFN synergistically augments the tumor suppressive effects of GEM on human iCCA cell growth. This effect of SFN is based on the inhibition of HDACs, leading to G2/M arrest; apoptosis; and suppressed invasion, migration, EMT, and angiogenesis. As a less-toxic phytochemical, SFN might eventually emerge as a viable modulator of GEM for patients with advanced iCCA.

## Figures and Tables

**Figure 1 cells-12-00687-f001:**
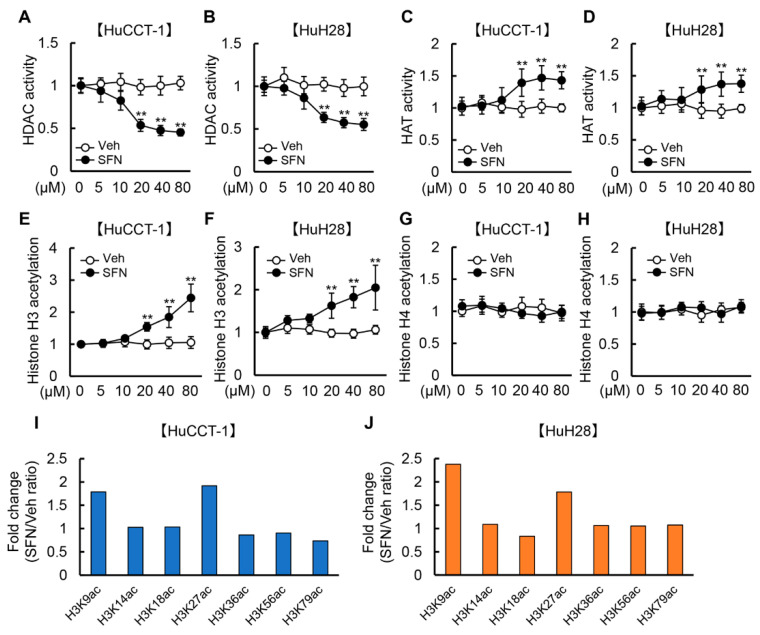
HDAC activity and histone H3 acetylation by treatment with SFN in iCCA cells. (**A**–**H**) HDAC activity (**A**,**B**), HAT activity (**C**,**D**), total histone H3 acetylation (**E**,**F**), and total histone H4 acetylation (**G**,**H**) in HuCCT-1 and HuH28 cells treated with SFN (0–80 μM). The values are shown as fold changes relative to 0 μM for each treatment group. Data are mean ± SD (*n* = 3 independent experiments with *n* = 8 samples per condition). ** *p* < 0.01 compared with the group treated with vehicle (Veh) at the same concentration. (**I**,**J**) Graphical analysis of the binding intensity of H3K9ac, H3K14ac, H3K18ac, H3K27ac, H3K36ac, H3K56ac, and H3K79ac in HuCCT-1 (**I**) and HuH28 (**J**) cells treated with SFN (20 μM) by using Histone H3 Peptide Array ELISA Kit. The values are shown as fold changes relative to Veh treatment at the same concentration.

**Figure 2 cells-12-00687-f002:**
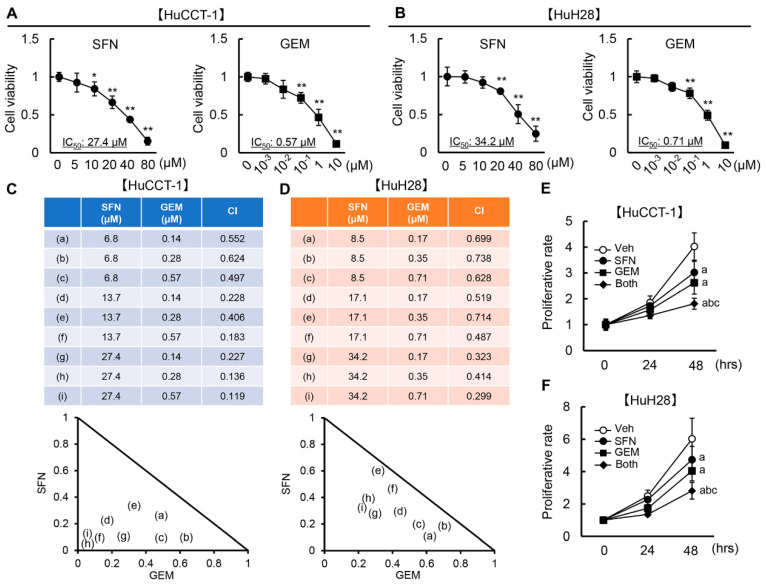
Cell viability and proliferation by treatment with SFN and GEM in iCCA cells. (**A**,**B**) Cell viability and IC50 in HuCCT–1 (**A**) and HuH28 (**B**) cells treated with SFN (0–80 μM) or GEM (0–10 μM). (**C**,**D**) The synergism of SFN and GEM on HuCCT–1 (**C**) and HuH28 (**D**) was evaluated by the combination index (CI) values. CI gives a quantitative definition of synergism (CI < 1), additive effect (CI = 1), and antagonism (CI > 1) (**E**,**F**). Cell proliferation of HuCCT–1 and HuH28 cells incubated with SFN and/or GEM at each IC50 for 0–48 hrs. The values are shown as fold changes relative to 0 μM for each treatment group (**A**,**B**) and the values at the start of each treatment (**E**,**F**). Data are mean ± SD (*n* = 3 independent experiments with *n* = 8 samples per condition) (**A**,**B**,**E**,**F**). * *p* < 0.05, ** *p* < 0.01 compared with the values of 0 μM for each treatment group (**A**,**B**). ^a^ *p* < 0.01, ^b^ *p* < 0.01, ^c^ *p* < 0.01 compared with the group treated for 48 h with Veh, SFN or GEM, respectively (**E**,**F**).

**Figure 3 cells-12-00687-f003:**
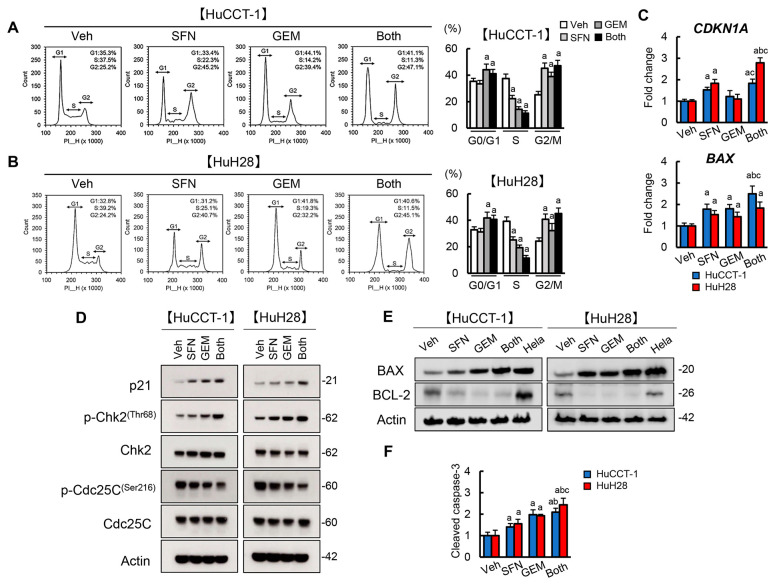
Cell cycle and apoptosis by treatment with SFN and GEM in iCCA cells. (**A**,**B**) Representative images of flow cytometric analysis for cell cycle distribution and percentages of cells at different cell cycle phases (G0/G1, S and G2/M) in HuCCT–1 (**A**) and HuH28 (**B**) cells treated with SFN and/or GEM. After the incubation with both agents for 12 h, cells were stained with propidium iodide (PI) and subjected to flow cytometry. (**C**) Relative mRNA levels of CDKN1A and BAX in HuCCT–1 and HuH28 cells. (**D**,**E**) Western blots for the markers related to G2/M arrest, p21, p–Chk2(Thr68) and p–Cdc25C(Ser216) (**D**), and the markers related to apoptosis, BAX and BCL–2 (**E**). (**F**) Cleaved caspase–3 level in HuCCT–1 or HuH28–cultured media. The mRNA expression levels were measured by quantitative RT-PCR (qRT–PCR), and *GAPDH* was used as an internal control for qRT–PCR (**C**). Actin was used as an internal control for western blotting (**D**,**E**). The values are shown as fold changes relative to the vehicle–treated group (Veh) (**C**,**F**). Data are mean ± SD (*n* = 3 independent experiments with *n* = 3 for A and B, *n* = 8 for **C**,**F**). ^a^ *p* < 0.01, ^b^ *p* < 0.01, ^c^ *p* < 0.01 compared with the group treated with Veh, SFN or GEM, respectively.

**Figure 4 cells-12-00687-f004:**
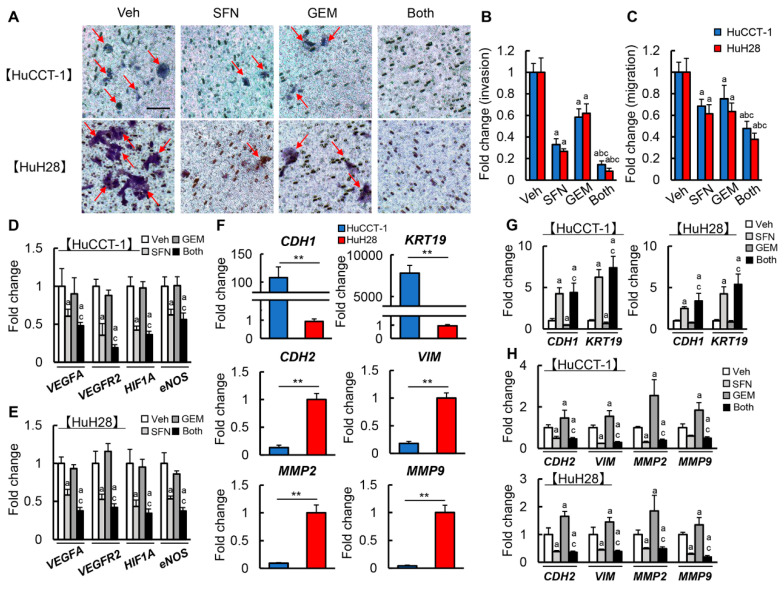
Cell invasion, pro-angiogenic property and EMT by treatment with SFN and GEM in iCCA cells. (**A**) Representative images of invasive HuCCT-1 and HuH28 cells treated with SFM and/or GEM. Scale bar; 50 μm. Red arrow; invasive cells. (**B**) Quantification of both cell lines invasion. (**C**) Quantification of both cell lines migration. (**D**,**E**) Relative mRNA expression of pro-angiogenic markers (VEGFA, VEGFR2, HIF1A, and NOS3) in HuCCT-1 and HuH28 treated with SFN and/or GEM. (**F**) Comparison between HuCCT-1 and HuH28 in the relative mRNA expression of the epithelial markers (CDH1 and KRT19) and mesenchymal markers (CDH2, VIM, MMP2 and MMP9) related to EMT. (**G**,**H**) Relative mRNA expression of epithelial markers (**G**) and mesenchymal markers (**H**) in HuCCT-1 and HuH28 treated with SFN and/or GEM. *GAPDH* was used as an internal control for qRT-PCR. The values are shown as fold changes relative to the vehicle-treated group (Veh) (**B**–**E**,**G**,**H**) or the values of the HuCCT-1 group (**F**). Data are mean ± SD (*n* = 3 independent experiments with *n* = 6 for B and *n* = 8 for B–G samples per condition); ^a^ *p* < 0.01, ^b^ *p* < 0.01, ^c^ *p* < 0.01 compared with the group treated with Veh, SFN or GEM, respectively (**B**–**E**,**G**,**H**). ** *p* < 0.01, indicating a significant difference between groups (**F**).

**Figure 5 cells-12-00687-f005:**
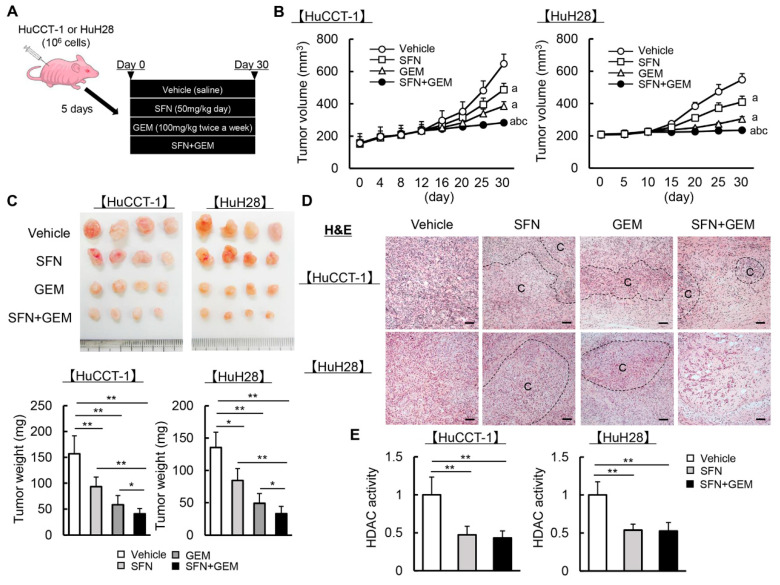
iCCA-derived xenograft tumor growth by treatment with SFN and GEM.(**A**) Experimental protocol. (**B**) Time course of HuCCT-1 and HuH28-grafted tumor volumes. (**C**) Representative images and weights of resected tumors at the end of the experiment. (**D**) Representative pictures of HuCCT-1 and HuH28-grafted subcutaneous tumors stained with H&E. C; cancerous lesions, Scale bar; 100 µm. (**E**) HDAC activity in the resected subcutaneous tumor tissues. The values are shown as fold changes to the vehicle-treated group (**E**). Data are mean ± SD (*n* = 20 tumors/10 mice; **B**,**C**,**E**). ^a^ *p* < 0.01, ^b^ *p* < 0.01, ^c^ *p* < 0.01 compared with the group treated with Veh, SFN or GEM, respectively at the end of the experiment (**B**). * *p* < 0.05, ** *p* < 0.01, indicating a significant difference between groups (**C**,**E**).

**Figure 6 cells-12-00687-f006:**
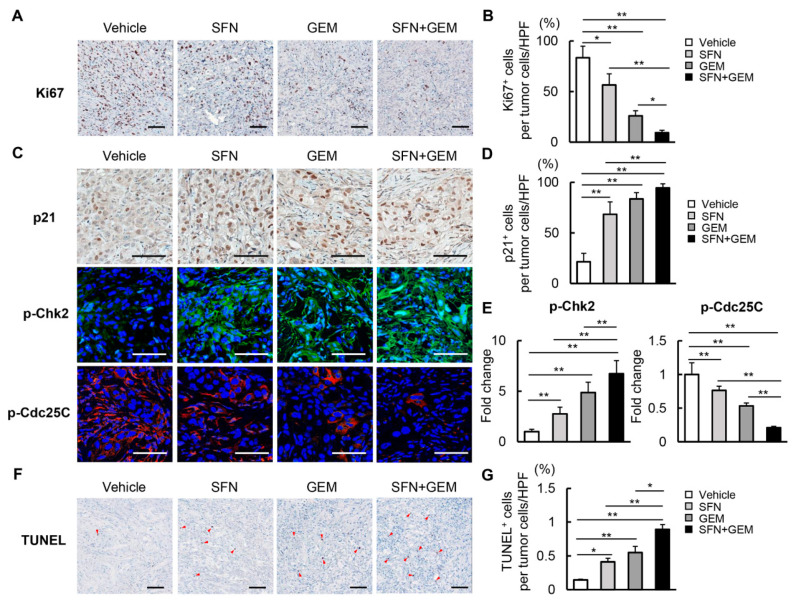
Cell proliferation and apoptosis in iCCA-derived xenograft tumors by treatment with SFN and GEM. (**A**,**C**,**F**) Representative images of HuCCT-1-grafted tumors stained with Ki67 (**A**), p21, p-Chk2, and p-Cdc25C (**C**), TUNEL (**F**). Red triangles indicate intratumor apoptotic cells. Scale bar; 100 µm. (**B**) Quantification of Ki67^+^ proliferative cancer cells. The values are indicated as Ki67^+^ cancer cells/total cancer cells (%) in high power field (HPF). (**D**) Quantification of p21^+^ cancer cells. Quantitative values are indicated as p21^+^ cancer cells/total cancer cells (%) in HPF. (**E**) Semi-quantitation of p-Chk2^+^ or p-CdC25C^+^ cancer cells in HPF. The values are shown as fold changes relative to the vehicle-treated group. (**G**) Quantification of TUNEL^+^ apoptotic cancer cells. The values are indicated as TUNEL^+^ cancer cells/total cancer cells (%) in HPF. Each quantitative analysis was performed for 10 fields per section. Data are mean ± SD (*n* = 20 tumors/10 mice; **B**,**D**,**E**,**G**). * *p* < 0.05, ** *p* < 0.01, indicating a significant difference between groups.

**Figure 7 cells-12-00687-f007:**
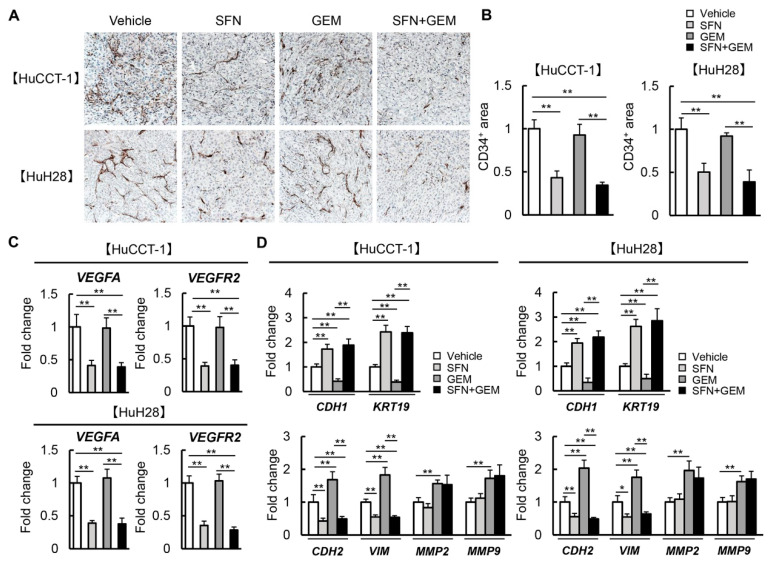
Intratumor angiogenesis and EMT in iCCA-derived xenograft tumors by treatment with SFN and GEM. (**A**) Representative images of CD34^+^ neovascularization in the HuCCT-1 and HuH28-grafted tumors. Scale bar; 100 µm. (**B**) Semi-quantitation of CD34^+^ vessels in the high-power field (HPF) by ImageJ software. Quantitative analysis included 10 fields per section. (**C**,**D**) Relative mRNA expression of pro-angiogenic VEGFA and VEGFR2 (**C**) and epithelial CDH1 and KRT19/mesenchymal CDH2, VIM, MMP2 and MMP9 (**D**) in the HuCCT-1 and HuH28-grafted subcutaneous tumors. *GAPDH* was used as an internal control for qRT-PCR. The values are shown as fold changes relative to the vehicle group (**B**–**D**). Data are mean ± SD (*n* = 20 tumors/10 mice). * *p* < 0.05; ** *p* < 0.01 indicating a significant difference between groups.

## Data Availability

The data that support the findings of this study are available from the corresponding author upon reasonable request.

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
