# Peer review of "Sulforaphane Potentiates Gemcitabine-Mediated Anti-Cancer Effects against Intrahepatic Cholangiocarcinoma by Inhibiting HDAC Activity"

_cells, 2023, doi:10.3390/cells12050687_

Round 1

Reviewer 1 Report

Thank you for submitting your work to Cells journal. I found the paper to be very thorough, scientifically exact, and also balanced and well written, congratulations. Other than the occasional typo, I have no major observations. While HDAC inhibition has somehow lost its appeal as a major cancer research direction in recent years (due to several failed PhII-II trials), SFN seems a viable chemotherapy modulator. As you mentioned in the Discussions, platinum agents (and also fluoropyrimidines) remain an important therapy option for iCCA, even in the I-O era. Their action depends greatly on chromatin structure and exposure (including epigenetic modulations) and as such were long considered as preferential partners for HDACis. Reproduction of your results with SFN added to a platinum-GEM chemotherapy combination would further confirm its potential clinical utility. Another interesting direction would be to search for a possible action pathway common to several/all malignant tumors. EMT seems particularly interesting, as it also has implications in the immune response.

Author Response

Dear Reviewer,

We really appreciate the reviewer for positively evaluating our manuscript and offering valuable advice.

Reviewer 2 Report

In the present scientific article entitled: “Sulforaphane potentiates gemcitabine-mediated anti-cancer effects against intrahepatic cholangiocarcinoma by inhibiting 3 HDAC activity”, the authors demonstrated the synergistic effect of the co-administration of gemcitabine and SFN in human iCCA cell lines, such as HuCCT-1 and HuH28.

In my opinion, this article, deserves the publication in Cells Journal, however I would like to suggest to the authors to make some adjustments for the publication.

Query#1

In the paragraph “Introduction”: the authors properly reported iCCA incidence and mortality, as well as they introduced the histone deacetylases (HDACs) proteins, which play a crucial role in the activation of genes involved in carcinogenesis, therefore HDAC inhibitors could represent a promising strategy in iCCA treatment.

According to this point, at page 2 line 60-62, the authors reported the following sentence: “Aberrant HDAC activity leads to diverse transcriptional gene regulation relevant to cancer cell differentiation, angiogenesis, proliferation, apoptosis, migration, and metastasis (8,9).”.

For a major clarity of the readers please mention two/ three genes expressed upon the activation of HDAC and involved in iCCA carcinogenesis.

Query#2

In the paragraph introduction the authors summarized the pharmacological properties of sulforaphane (SFN), however in my opinion is not clear at line 71, what the authors would mine stating that the SFN is able to induce phase II reaction metabolism and therefore possess chemopreventive properties in a wide range of cancer. I kindly ask to the authors to clarify this point.

Query#3

In the results the authors clearly demonstrated the synergistic effect of the co-administration of gemcitabine and SFN in human iCCA cell lines, such as HuCCT-1 and HuH28. The drugs combination significantly mediated growth inhibition, as well as blocked the cell cycle in G2/M phase, also a reduction of invasion in CCA cancer cell lines to less than 20% compared to the control was observed. I really appreciate the extensive work made by the authors, indeed is important to find novel combinatory treatments with standard chemotherapeutic drugs such as the gemcitabine. Indeed, this approach was also employed for other types of cancer such us, in the high aggressive diffuse malignant peritoneal mesothelioma (DMPM) and in pancreatic ductal adenocarcinoma (PDAC).

Therefore, I suggest to the authors to check the following papers and to add in the introduction a brief overview of the therapeutic efficacy of combination therapy with gemcitabine in other types of cancer citing these updated literature:

1)    Li Petri, G., Pecoraro, C., Randazzo, O., Zoppi, S., Cascioferro, S. M., Parrino, B., Carbone, D., El Hassouni, B., Cavazzoni, A., Zaffaroni, N., Cirrincione, G., Diana, P., Peters, G. J., & Giovannetti, E. (2020). New Imidazo[2,1-b][1,3,4]Thiadiazole Derivatives Inhibit FAK Phosphorylation and Potentiate the Antiproliferative Effects of Gemcitabine Through Modulation of the Human Equilibrative Nucleoside Transporter-1 in Peritoneal Mesothelioma. Anticancer research40(9), 4913–4919. https://doi.org/10.21873/anticanres.14494

2)    Miller, A. L., Garcia, P. L., & Yoon, K. J. (2020). Developing effective combination therapy for pancreatic cancer: An overview. Pharmacological research155, 104740. https://doi.org/10.1016/j.phrs.2020.104740

Author Response

Dear Reviewer,

We really appreciate the reviewer for positively evaluating our manuscript and offering valuable advice.

Query#1

In the paragraph “Introduction”: the authors properly reported iCCA incidence and mortality, as well as they introduced the histone deacetylases (HDACs) proteins, which play a crucial role in the activation of genes involved in carcinogenesis, therefore HDAC inhibitors could represent a promising strategy in iCCA treatment.

According to this point, at page 2 line 60-62, the authors reported the following sentence: “Aberrant HDAC activity leads to diverse transcriptional gene regulation relevant to cancer cell differentiation, angiogenesis, proliferation, apoptosis, migration, and metastasis (8,9).”.

For a major clarity of the readers please mention two/ three genes expressed upon the activation of HDAC and involved in iCCA carcinogenesis.

Answer

We really appreciate the reviewer for giving the important suggestion. According to the reviewer’s comment, we additively introduced two/ three genes expressed upon the activation of HDAC and involved in iCCA carcinogenesis as below.

HDAC2 depletion activates apoptosis via p53 and Bax activation and Bcl2 suppression induces cell-cycle arrest by induction of p21 and suppression of cyclin E2, cyclin D1, and CDK2 (J Cell Biochem. 2012;113(6):2167-77). Morine et al. have reported that intratumor HDAC expression is positively correlated with HIF-1α a stimulus factor for local hypoxia and increased angiogenesis in resected iCCA tissues (Surgery. 2012;151(3):412–9.).

Query#2

In the paragraph introduction the authors summarized the pharmacological properties of sulforaphane (SFN), however in my opinion is not clear at line 71, what the authors would mine stating that the SFN is able to induce phase II reaction metabolism and therefore possess chemopreventive properties in a wide range of cancer. I kindly ask to the authors to clarify this point.

Answer

We really apologize the reviewer for inappropriate description. Although the induction of phase II reaction metabolism is one of the mechanisms of SFN-mediated anti-cancer property, this pharmacological mechanism is different from the argument of our present study; the inhibition of HDAC. Because this part is confusing for the reader to understand, we eliminated this part in the revised manuscript.

Query#3

In the results the authors clearly demonstrated the synergistic effect of the co-administration of gemcitabine and SFN in human iCCA cell lines, such as HuCCT-1 and HuH28. The drugs combination significantly mediated growth inhibition, as well as blocked the cell cycle in G2/M phase, also a reduction of invasion in CCA cancer cell lines to less than 20% compared to the control was observed. I really appreciate the extensive work made by the authors, indeed is important to find novel combinatory treatments with standard chemotherapeutic drugs such as the gemcitabine. Indeed, this approach was also employed for other types of cancer such us, in the high aggressive diffuse malignant peritoneal mesothelioma (DMPM) and in pancreatic ductal adenocarcinoma (PDAC).

Therefore, I suggest to the authors to check the following papers and to add in the introduction a brief overview of the therapeutic efficacy of combination therapy with gemcitabine in other types of cancer citing these updated literature:

1)    Li Petri, G., Pecoraro, C., Randazzo, O., Zoppi, S., Cascioferro, S. M., Parrino, B., Carbone, D., El Hassouni, B., Cavazzoni, A., Zaffaroni, N., Cirrincione, G., Diana, P., Peters, G. J., & Giovannetti, E. (2020). New Imidazo[2,1-b][1,3,4]Thiadiazole Derivatives Inhibit FAK Phosphorylation and Potentiate the Antiproliferative Effects of Gemcitabine Through Modulation of the Human Equilibrative Nucleoside Transporter-1 in Peritoneal Mesothelioma. Anticancer research, 40(9), 4913–4919. https://doi.org/10.21873/anticanres.14494

2)    Miller, A. L., Garcia, P. L., & Yoon, K. J. (2020). Developing effective combination therapy for pancreatic cancer: An overview. Pharmacological research, 155, 104740. https://doi.org/10.1016/j.phrs.2020.104740

Answer

We really appreciate the reviewer’s suggestion. According to the reviewer’s suggestion, we explained the GEM-based combination therapy with other agents against other cancer cells in the Introduction.

Reviewer 3 Report

The paper showed that the role of SFN in iCCA. The author suggest SFN inhibit HDAC activity. Furthermore, it function as anti-proliferative, pro-apoptotic, anti-invasive, anti-EMT and anti-angiogenic effector. This is a valuable submission that I recommend for publication with a few of changes.  

Comments:

(1) I can't find Figure 8 mentioned in Discussion.

(2) In figure 4, a migration assay in iCCA cells is also required in addition to a Matrigel invasion assay.

(3) In Figure 7, it is necessary to confirm the expression at the protein level of Ecadherin, Ncadherin, vimentin, MMP2 and MMP9 in human iCCA-derived xenograft tumors.

(4) There are some minor language errors. The authors should be revised the manuscript with an English language editor to make it more readable.

Author Response

Dear Reviewer,

We really appreciate the reviewer for positively evaluating our manuscript and offering valuable advice.

The paper showed that the role of SFN in iCCA. The author suggest SFN inhibit HDAC activity. Furthermore, it function as anti-proliferative, pro-apoptotic, anti-invasive, anti-EMT and anti-angiogenic effector. This is a valuable submission that I recommend for publication with a few of changes. 

Comments:

I can't find Figure 8 mentioned in Discussion.

Answer

We apologize the reviewer for incorrect description. We showed Graphical Abstract instead of Figure 8 in this manuscript. We omitted this part.

In figure 4, a migration assay in iCCA cells is also required in addition to a Matrigel invasion assay.

Answer

We appreciate the reviewer for important suggestion. According to the reviewer’s suggestion, we newly performed a set of experiment for cell migration using A CytoselectTM 96-well Cell Migration Assay (8μm) (Cell Biolabs Inc., San Diego CA, USA). As shown in new Figure 4C, cell migration was also suppressed by treatment with SFN or GEM in both iCCA cells. Moreover, combination treatment enhanced the suppressive effect of each single agent. We added this result in the revised manuscript.

In Figure 7, it is necessary to confirm the expression at the protein level of Ecadherin, Ncadherin, vimentin, MMP2 and MMP9 in human iCCA-derived xenograft tumors.

Answer

We really thank the reviewer for pointing out important issues. As the reviewer mentioned, we measured above protein levels in tumor tissue samples by using each ELISA assay. As shown in new Figure S5A, E-cadherin levels in human iCCA-derived xenograft tumors were increased by treatment with SFN while decreased by treatment with GEM as compared to vehicle group. By contrast, intratumor levels of N-Cadherin, Vimentin, MMP-2 and MMP-9 were decreased by treatment with SFN while increased by treatment with GEM (Figure S5B). We added these results in the revised manuscript.

There are some minor language errors. The authors should be revised the manuscript with an English language editor to make it more readable.

Answer

We appreciate for the reviewer for finding minor language errors. We corrected these errors.

Reviewer 4 Report

The effects of combining sulforaphane (SFN) and gemcitabine (GEM) on the proliferation of human intrahepatic cholangiocarcinoma (iCCA) cells were investigated in the study by Fumimasa et al. Results showed that SFN elevated histone H3 acetylation in iCCA cells while reducing overall histone deacetylase (HDAC) activity. By triggering cell cycle arrest and apoptosis, impeding cell invasion and angiogenesis, and inhibiting the epithelial-mesenchymal transition, SFN also enhanced the effects of GEM (EMT).  The combination of SFN and GEM prevented the development of subcutaneous tumor grown from human iCCA cells in xenograft tumor growth experiment, enhanced apoptosis, and reduced angiogenesis. 

Comments-

1- The authors used CCA cell lines (HuCCT1 and Huh28) Additionally, suggest using normal/primary cell lines, which may improve in our understanding of new and effective therapeutics for this disease.

2- Histone H3 acetylation in iCCA cells in figure illustrates the HDAC activity of SFN. Please specify which histones are acetylated with SFN treatment (i.e. H3, H4...)

3- The authors need to use good quality images in Figure 4 A.

Author Response

Dear Reviewer,

We really appreciate the reviewer for positively evaluating our manuscript and offering valuable advice.

The effects of combining sulforaphane (SFN) and gemcitabine (GEM) on the proliferation of human intrahepatic cholangiocarcinoma (iCCA) cells were investigated in the study by Fumimasa et al. Results showed that SFN elevated histone H3 acetylation in iCCA cells while reducing overall histone deacetylase (HDAC) activity. By triggering cell cycle arrest and apoptosis, impeding cell invasion and angiogenesis, and inhibiting the epithelialmesenchymal transition, SFN also enhanced the effects of GEM (EMT).  The combination of SFN and GEM prevented the development of a subcutaneous tumor grown from human iCCA cells in a xenograft tumor growth experiment, enhanced apoptosis, and reduced angiogenesis. 

Comments

1- The authors used CCA cell lines (HuCCT1 and Huh28) Additionally, I suggest using normal/primary cell lines, which may improve in our understanding of new and effective therapeutics for this disease.

Answer

We really appreciate the reviewer’s kind comments. As the reviewer’s suggestion, it is quite important to evaluate whether these agents toxically affect normal/primary biliary epithelial cells. Actually, we have these data using Human Intrahepatic Biliary Epithelial Cells (HIBEpiC) (ScienCell Research Laboratories, Inc.). As shown in new Figure S1B and S1C, SFN did not change HDAC activity and cell viability in cultured HIBEpiC. Moreover, GEM did not reduce cell viability of cultured HIBEpiC. These results suggest that the combination regimen of both agents would be available without biliary damage. We added these results in the revised manuscript.

2- Histone H3 acetylation in iCCA cells in figure 1 illustrates the HDAC activity of SFN. Please specify which histones are acetylated with SFN treatment (i.e. H3, H4...)

Answer

We thank the reviewer for important suggestion. Histone acetylation was occurred in histone H3 and H4. Thus, according to the reviewer’s comment, we newly evaluated the effect of sulforaphane on histone H4 acetylation by using EpiQuik™ Total Histone H4 Acetylation Detection Fast Kit (Epigentek). As shown in new Figure 1H and 1G, treatment with sulforaphane did not modify histone H4 acetylation in both HuCCT-1 and HuH28 cell. In the current study, we did not identify pharmacological mechanism to explain the differential effect of sulforaphane on H3 and H4 acetylation. Thus, further analysis would be needed to clarify this point. We added these results and discussion in the revised manuscript.

3- The authors need to use good quality images in Figure 4 A.

Answer

We apologize the reviewer for displaying unclear pictures in Figure 4A. We replaced these pictures.

Round 2

Reviewer 4 Report

The Authors have addressed  all of my comments. Now I am satisfied with this manuscript.